# Glomerular Filtration Rate in Former Extreme Low Birth Weight Infants over the Full Pediatric Age Range: A Pooled Analysis

**DOI:** 10.3390/ijerph17062144

**Published:** 2020-03-24

**Authors:** Elise Goetschalckx, Djalila Mekahli, Elena Levtchenko, Karel Allegaert

**Affiliations:** 1Department of Development and Regeneration, KU Leuven, Herestraat 49, 3000 Leuven, Belgium; elise.goetschalckx@student.kuleuven.be (E.G.); djalila.mekahli@uzleuven.be (D.M.); elena.levtchenko@uzleuven.be (E.L.); 2Department of Pediatric Nephrology and Organ Transplantation, University Hospitals Leuven, Herestraat 49, 3000 Leuven, Belgium; 3Department of Pharmacy and Pharmaceutical Sciences, KU Leuven, Herestraat 49, 3000 Leuven, Belgium; 4Department of Clinical Pharmacy, Wytemaweg Hospital Pharmacy Postbus 2040, Erasmus MC, Rotterdam, The Netherlands

**Keywords:** glomerular filtration rate, Brenner hypothesis, extreme low birth weight infants, renal outcome

## Abstract

Various cohort studies document a lower glomerular filtration rate (GFR) in former extremely low birth weight (ELBW, <1000 g) neonates throughout childhood when compared to term controls. The current aim is to pool these studies to describe the GFR pattern over the pediatric age range. To do so, we conducted a systematic review on studies reporting on GFR measurements in former ELBW cases while GFR data of healthy age-matched controls included in these studies were co-collected. Based on 248 hits, 6 case-control and 3 cohort studies were identified, with 444 GFR measurements in 380 former ELBW cases (median age 5.3–20.7 years). The majority were small (17–78 cases) single center studies, with heterogeneity in GFR measurement (inulin, cystatin C or creatinine estimated GFR formulae) tools. Despite this, the median GFR (mL/min/1.73 m^2^) within case-control studies was consistently lower (−13%, range −8% to −25%) in cases, so that a relevant minority (15–30%) has a eGFR<90 mL/min/1.73 m^2^). Consequently, this pooled analysis describes a consistent pattern of reduced eGFR in former ELBW cases throughout childhood. Research should focus on perinatal risk factors for impaired GFR and long-term outcome, but is hampered by single center cohorts, study size and heterogeneity of GFR assessment tools.

## 1. Introduction

Due to improvements in perinatal healthcare, the mortality of preterm neonates decreased over the last decades. This decrease applies even more to infants born with an extremely low birth weight (ELBW), i.e., a birth weight <1000 g [1,2]. However, there is growing evidence that preterm birth in itself is a trait for life as it has consequences on different organ systems, including the renal system. Glomerulogenesis normally finishes at 34–36 weeks of fetal life, but this program is disturbed after preterm birth by preterm birth itself, as well as perinatal events, so that former preterm infants have a decreased radial glomerular count compared to term cases [3,4]. Brenner et al. has put forward the concept (‘Brenner hypothesis’) that nephron endowment and subsequent single nephron hyperfiltration is a significant factor in the pathogenesis of subsequent chronic kidney disease (CKD) and hypertension [3,4,5]. As preterm birth disturbs glomerulogenesis and nephron endowment, it is a risk factor for CKD and it may be useful to follow up the renal function in former ELBW children, with glomerular filtration rate (GFR) as the most obvious indicator of renal function [3,4,5].

As we know, GFR can be measured or estimated. GFR measurement by exogenous markers such as inulin, iohexol, mannitol, Cr-EDTA or iothalamate is still the gold standard, but this has its burdens and limitations as screening tool [6,7]. Therefore, GFR is more commonly estimated (eGFR) by endogenous markers like serum creatinine (SCr) or cystatin C (CysC). The SCr-based GFR equation is hereby mostly used, with the CysC-based GFR equations as valid alternatives. The combination of both (SCr-CysC-based GFR equation) has a higher accuracy and precision, also in children [8]. The assessment of eGFR in former ELBW children throughout childhood has reported in observational studies, sometimes compared to age-matched controls. The current aim is to pool these studies to describe the GFR trend over the pediatric age range in former ELBW cases.

## 2. Materials and Methods

This systematic review was conducted along the PRISMA (Preferred Reporting Items for Systematic Reviews and Meta-Analyses) guidelines [9]. To be eligible, participants in the included studies had to be former ELBW newborns, with an age range from infancy until young adulthood and GFR outcome data. Healthy age-matched peers (born term with a normal birth weight) included in these studies as controls were co-enrolled in this analysis to facilitate comparison of GFR between these cases and controls.

The search was performed on March 16th, 2019 without language restriction to retrieve all relevant articles searching Pubmed, Embase, Web of Science and ClinicalTrials.gov respectively [**Pubmed**: (“infant, extremely low birth weight”[Mesh] OR elbw[tiab] OR extremely-low-birth-weight[tiab] OR extremely-preterm[tiab] OR extremely-low-birthweight[tiab]) AND (“kidney function tests”[Mesh] OR “cystatin C”[Mesh] OR “creatinine”[Mesh] OR “glomerular filtration rate”[Mesh] OR “renal insufficiency, chronic”[Mesh] OR renal-function[tiab] OR renal-complications[tiab] OR renal-follow-up[tiab] OR renal-outcome[tiab] OR kidney-function[tiab]); **Embase:** (‘extremely low birth weight’:ti,ab,kw OR ‘extremely preterm infant’:ti,ab,kw) AND (‘kidney function’:ti,ab,kw OR ‘kidney function test’:ti,ab,kw OR ‘cystatin c’:ti,ab,kw OR ‘creatinine’:ti,ab,kw OR ‘glomerulus filtration rate’:ti,ab,kw OR ‘kidney failure’:ti,ab,kw OR ‘chronic kidney failure’:ti,ab,kw); **Web of Science**: (“infant, extremely low birth weight” OR elbw OR “extremely low birth weight” OR “extremely preterm” OR “extremely low birthweight”) AND (“kidney function tests” OR “cystatin C” OR creatinine OR “glomerular filtration rate” OR “Renal Insufficiency, Chronic” OR “renal funct*” OR “renal complicatio*” OR “renal follow” OR “renal outcome”); **ClinicalTrials.gov**: (extremely low birth weight OR elbw OR extremely low birthweight OR extremely preterm) AND (kidney function test OR cystatin C OR creatinine OR glomerular filtration rate OR GFR OR chronic kidney disease OR CKD OR renal function OR renal complication OR renal follow OR renal outcome)].

Articles were screened based on title, abstract and finally full text containing information about participants (specific ELBW children, with a GFR assessment), while studies including but not reporting specific on GFR outcome in ELBW cases were not included) by the first author (EG), and uncertainties were discussed with the final author (KA) until agreement. For the full papers retained in the search, the reference list and list of citing articles were screened.

The STROBE statement checklist (Strengthening the Reporting of Observational Studies in Epidemiology) to evaluate the quality of the obtained articles and to assess the risk of bias (cohort, case-control) were used [10]. Additional quality assessment was performed, using the Newcastle–Ottawa assessment scale for case-control or cohort studies respectively [11].

From the individual papers, the following data were extracted and tabulated: study design, study size, year of birth, current age of the patient population, birth weight of the patient population, outcome parameters and results (also others than GFR), exclusion criteria, GFR measurement (including technique and/or estimation formula used) summary of reported findings and we searched if the researchers had adjusted for covariates. The mean/median GFR or eGFR values mentioned in the included articles were extracted to plot the GFR values over the pediatric age range. Data extraction was performed by the first author (EG), verified by the final author (KA) and differences were discussed until agreement.

## 3. Results

Following the search strategy described, 248 (232 + 16) hits were analyzed to result in 6 case-control and 3 ELBW cohort studies for the analysis (Figure 1). The key characteristics of the individual case-control studies are provided in Table 1 [12,13,14,15,16,17,18], cohort studies in Table 2 [19,20,21].

One study (Krakow cohort) collected Cys C values in the same cohort at the age of 7 and 11 years and calculated eGFR values in the latest paper [14,18]. To provide an overview on the characteristics, data (eGFR data calculated) reported at the age of 7 years were added to Table 1 (*italics*) [18].

Table 2 provides the same information for the 3 cohort studies retained in the analysis [19,20,21].

In total, 444 GFR or eGFR measurements in 380 ELBW cases were retrieved, with an age range (median values in the different cohorts) between 5.3 and 20.7 years. The majority of studies were single center, small (17 to 78 cases), with heterogeneity in GFR measurement (inulin, cystatin C or creatinine) and estimated GFR formulae (overview Appendix A) tools. Despite this, the GFR (mL/min/1.73 m^2^) was consistently lower in cases (median −13%, range, −8% to −25%) without significant trend over pediatric life. The median decrease (−13%) reflects a shift of eGFR estimates of about 0.5 to 1 SD, so that a relevant minority (15–30%) of former ELBW cases has an eGFR <90 mL/min/1.73 m^2^.

Besides the GFR values, data on other aspects of renal and general outcome (like renal ultrasound, renal tubular functions, blood pressure, biometry and body composition) were co-reported. Some papers aimed to explore perinatal risk factors like birth weight, small for gestational age (SGA), perinatal growth, Apgar score, indomethacin/ibuprofen or steroid use or gestational age, but any firm conclusion on these risk factors remained hampered by the cohort sizes.

Quality assessment based on the case-control and cohort study STROBE checklist resulted in moderate to high scores of the nine observational studies retained in the analysis. Five of the papers explicitly discussed risks of biases in the recruitment strategy [13,15,16,17,20]. In many studies, it was not entirely clear how patient selection and follow-up have been conducted. The assessment of these studies using the Newcastle–Ottawa scale (selection, comparability, outcome) also suggests moderate to high scores (Table 3).

## 4. Discussion

Using a systematic review on cohort studies reporting on GFR measurements in 380 former ELBW cases throughout pediatric life, the eGFR (mL/min/1.73 m^2^) was consistently lower in cases (median −13%, range, −8% to −25%). The median decrease (−13%) is similar to a shift of eGFR estimates of about 0.5 to 1 SD, so that a relevant minority (15–30%) of former ELBW cases has an eGFR <90 mL/min/1.73 m^2^ [16,19,21]. We hereby confirmed the pattern of reduced GFR in former ELBW cases throughout pediatric age in line with the Brenner hypothesis [4,5].

To further explore the extent of the relative difference in GFR and potential trends throughout pediatric life, we plotted the mean or median (estimated) GFR in the 6 case-control studies in Figure 2. As one of the cohorts has been assessed twice, we have added arrows to link these specific case-control GFR estimates. Based on visual inspection, the percent GFR (median −13%, range, −8% to −25%) showed no clear trend of percent difference in eGFR over age between cases and controls, but this interpretation is hampered by the use of median cohort eGFR data, and not the individual eGFR data as collected.

Consequently, based on the impaired GFR outcome, clinical research should further explore the perinatal risk factors associated with this outcome and call for action to further explore the subsequent relevance of these findings beyond pediatric life. However, there are study-related issues that make such an effort difficult.

Despite the fact that the quality assessment of the studies resulted in moderate to high scores (Table 3), we still observed major relevant burdens to facilitate effective pooling. First, the source documents did not always provided sufficient information to ensure ‘full’ inclusions of all ELBW cases, and this may have resulted in additional overrepresentation in SGA cases if only the SGA subgroup could be included in the current pooling effort [13,17,19]. An obvious next step is to pool based on the individual data. Such an effort should also facilitate the exploration of the association between perinatal characteristics (like birth weight, small for gestational age (SGA), Apgar score, indomethacin/ibuprofen or steroid use or gestational age) and GFR outcome, similar to the effort recently reported to pool the ELBW renal outcome data at the age of 11 years [22]. Second and at least as relevant, such pooling effort needs to consider the heterogeneity in GFR measurement (inulin, cystatin C or creatinine estimated GFR formulae) tools and assays (cystatin C, creatinine) used [6,22].

## 5. Conclusions

Despites the study related limitations to enable pooling (small size, single center, heterogeneity GFR measurements), the pattern of reduced GFR in former ELBW cases throughout pediatric age has been confirmed. The GFR (mL/min/1.73 m^2^) was consistently lower in cases (median −13%, range, −8% to −25%), similar to a shift of eGFR estimates of about 0.5 to 1 SD, so that a relevant minority (15–30%) of former ELBW cases has a eGFR <90 mL/min/1.73 m^2^. Consequently, we should further explore the perinatal risk factors associated with impaired GFR outcome and the subsequent relevance of these findings beyond pediatric life.

## Figures and Tables

**Figure 1 ijerph-17-02144-f001:**
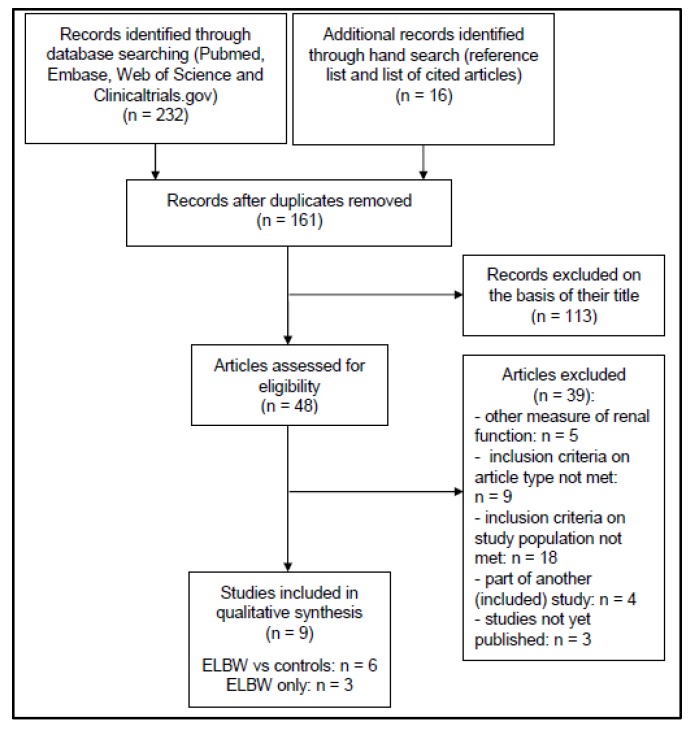
Preferred Reporting Items for Systematic Reviews and Meta-Analyses (PRISMA) flow chart describing the search strategy and results.

**Figure 2 ijerph-17-02144-f002:**
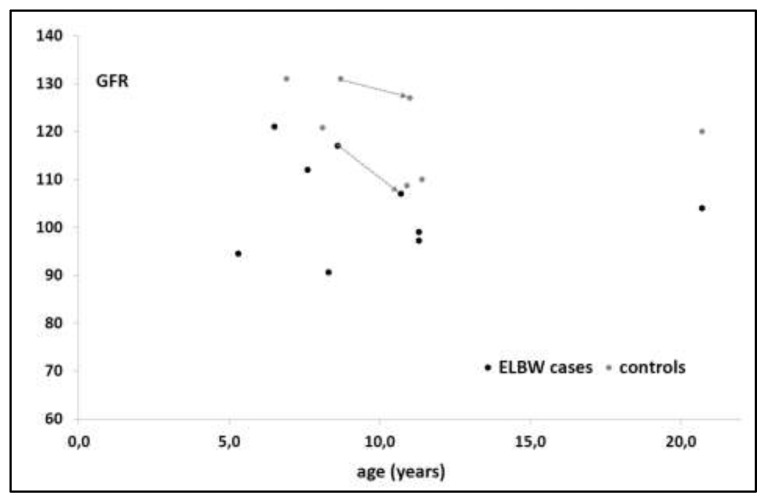
Mean or median (estimated) glomerular filtration rate (eGFR) in the 6 case-control studies over age (years) [12,13,14,15,16,17]. The arrows connect the same cohort assessed at the age of 7 and 11 years respectively [17,18] (ELBW: extreme low birth weight).

**Table 1 ijerph-17-02144-t001:** Key characteristics and findings of the 6 case-control cohort studies included (chronological) [12,13,14,15,16,17].

Study	Number, Age	Assay, Formula	Results	Comments
Rodriguez-Soriano 2005 [12]	40 cases (8.6, 6.1–12.4), 43 controls (8.5, 5.2–13) y.	modified Jaffé crea, with subsequent eGFR Schwartz.	cases vs. controls: 117 (17, range 86–152) vs. 131 (17, range 97–173) mL/min/1.73 m^2^	single center, 40/75 ELBW cases included. Controls minor surgery cases.
Keijzer-Veen 2007 [13]	23/52 ELBW-SGA cases (20.7, SD 0.3); 30 controls (20.7, SD 0.8) y.	inulin clearance, at baseline and stimulated (protein rich lunch + low (2 µg/kg/min dopa).	cases vs. controls (baseline/stimulated): 104, SD 17/116, SD 27 vs. 120, SD 28 to 141, SD 34 mL/min; 107, SD 15 to 119, SD 23 vs. 112, SD 22, to 131, SD 26 mL/min/1.73 m^2^	cases recruited from a population follow-up study former preterms (the Netherlands); controls volunteers.
Kwinta 2011 [18]	78 cases, mean age 6.5; 38 controls (6.9) y.	Cys C, nephelometric assay (ref value 4–10 y 0.53–0.95 mg/L).	cases vs. controls (7 y): Cys C 0.64 (0.07) vs. 0.59 (0.07) mg/L. Using the Hoek formula, equal to 121 vs 131 mL/min/1.73 m^2^.	single center, 78/89 cases included, controls from general practitioners’ offices.
Starzec 2016 [14]	64/78 cases re-studied at 10.7; 36/38 controls at 11 y.	Cys C + eGFR, based on Hoek formula; crea (assay ?)	cases vs. controls (11 y): Cys C 0.72 (SD 0.15) vs. 0.61 (SD 0.08) mg/L; eGFR 107.3 to 127.4 mL/min/1.73 m^2^; crea 43.2, SD 7.7 vs. 46.3, SD 7.6 µmol/L)	single center, 64/78 cases retained, controls from general practitioner offices.
Yamamura-Miyazaki 2017 [15]	48 cases, mean age 8.3 y, and 48 controls, 8.1 y.	Cys C, latex turbidimetry; Cys-eGFR (Uemura formula); Crea enzymatic; Crea-eGFR (assay ?)	Cases vs. controls: Cys C 1.08 (0.17) vs. 0.82 (0.09) mg/L, Cys-eGFR 90.6 (15.5) vs. 120.8 (14.5) mL/min/1.73 m^2^; crea 0.46 (0.09) vs. 0.37 (0.08) mg/dl; crea-eGFR 95.4 (15.5) vs. 123.9 (14.5) mL/min/1.73 m^2^	single center, 48/86 cases included; controls were outpatient clinics cases.
Raaij-makers 2017 [16]	59 cases (11.3, SD 1.4); 71 controls (10.9, SD 1.3) y.	Cys C (turbidimetry), Cys-eGFR (CAPA formula) Crea (enzymatic), Crea-eGFR (Schwartz).	Cases vs. controls: Cys C 0.96 (0.12) vs. 0.87 (0.11) mg/L; Cys-eGFR 97.2 (13.6) vs. 108.7 (15.3) mL/min/1.73 m^2^; Crea 0.57 (0.1) vs. 0.56 (0.08) mg/dl; Crea-eGFR 111 (17) vs. 111 (15) mL/min/1.73 m^2^.	single center, 93/140 cases, but blood sampling in only 59 cases. Controls were volunteers.
Vollsaeter 2018 [17]	17 SGA cases (mean 11.3), and 45 controls (11.4) y.	Cys C (immuno-maldi); Crea (chromatography); eGFR Schwartz, Gao (crea), Zappitelli (crea+Cys C)	Cases vs. controls: Cys C 0.91 vs 0.86 mg/L; Cys-GFR; Crea 53.6 vs. 51 µmol/L; GFR_Schw_ 99 vs. 110; GFR_Gao_ 98.4 va 105.6; GFR_Zapp_ 95.1 vs. 104.8 mL/min/1.73 m^2^.	regional cohort with 17 SGA-ELBW cases. Controls volunteers from same maternity.

ELBW: extreme low birth weight; BUN: blood urea nitrogen; SGA: small for gestational age; eGFR: (estimated) glomerular filtration rate; Cys C: cystatin C; crea: creatinine; y: years.

**Table 2 ijerph-17-02144-t002:** Key characteristics of the 3 cohort studies included (chronologically) [19,20,21].

Study	Number, Age	Assay, Formula	Results	Comments
Bacchetta 2009 [19]	50 cases, 7.6 (range 5.8–10.3) y.	inulin clearance	average GFR 112 (range 91–158 mL/min/1.73 m^2^)	single center study, 50/143 with GFR in 46/50 cases, 39 ELBW cases
Zaffanello 2010 [20]	26 ELBW cases. 5.3 (95% CI 5.2–6.3) y.	Cys C assay (nephelometry); crea (modified Jaffé); Schwartz (crea, crea/Cys-C/BUN)	Median Cys C 0.67 mg/L; crea 0.42 mg/dl; Schwartz_crea_: 109 mL/min/1.73 m^2^; Schwartz_crea-CysC-BUN_: 94.5 mL/min/1.73 m^2^	single center study, recruited 69/97 contacted cases, but 1000–1500 g birth weight cases also recruited.
Matsumura 2019 [21]	43 cases with follow-up, 7 (range 2–22) y.	Crea, assay unknown; eGFR Japanese children.	only qualitative reporting: 12 (28%) had low GFR (<90 mL/min/1.73 m^2^).	single center, retrospective, cross sectional study.

ELBW: extreme low birth weight; BUN: blood urea nitrogen; SGA: small for gestational age; eGFR: (estimated) glomerular filtration rate; Cys C: cystatin C; crea: creatinine; y: years.

**Table 3 ijerph-17-02144-t003:** The Newcastle–Ottawa quality assessment scale applied to the studies retained in this analysis. Questions related to selection (Q1−4), comparability (Q5) and outcome (Q6–Q8) [11].

**Questions Q1–Q8 Case-Control**	**Vollsaeter 2018 [17]**	**Raaijmakers 2017 [16]**	**Yamamura-Miyazaki 2017 [15]**	**Starzec 2016 [14]**	**Keyzer-Veen 2007 [13]**	**Rodriguez-Soriano 2005 [12]**
case definition adequate	+	+	+	+	+	+
representativeness cases	+	+	+	+	unknown	+
selection controls	+	+	outpatients	outpatients	+	minor surgery
definition controls	+	+	outpatients	outpatients	+	minor surgery
comparability cases-controls	+	+	+	+	+	+
ascertainment exposure	+	+	+	+	+	+
ascertainment method	+	+	+	+	+	+
response rate *	+(93%)	+(66%)	+(55%)	+(82%)	unknown	+(84%)
**questions Q1–Q8 case cohort**	**Matsumura 2019 [21]**	**Zaffanello 2010 [20]**	**Bachetta 2009 [19]**			
representativeness	+	+	+			
selection non-exposed	n.a.	n.a.	n.a.			
ascertainment exposure	+	+	+			
outcome of interest presence	+	+	+			
comparability	+	+	+			
outcome assessment	+	+	+			
follow-up, period	+	+	+			
follow-up, adequacy	+(81%)	+(71%)	+(35%)			

* = response rate as mentioned in the source document, for cases (n.a.: not applicable).

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
