# Peer review of "Glomerular Filtration Rate in Former Extreme Low Birth Weight Infants over the Full Pediatric Age Range: A Pooled Analysis"

_ijerph, 2020, doi:10.3390/ijerph17062144_

Round 1

Reviewer 1 Report

Authors addressed and answered all concerns raised by the reviewers.

Reviewer 2 Report

The authors have adequately dealt with my criticisms. 

This manuscript is a resubmission of an earlier submission. The following is a list of the peer review reports and author responses from that submission.

Round 1

Reviewer 1 Report

This paper addresses the reduction in kidney function in infants with a formerly extreme low birth weight. Despite the limitations of the analyses, acknowledged by the authors, the conclusion is that kidney function in these infants is reduced by approximately 15% compared to controls.

While these data are seemingly clear, the authors should be a bit more careful with their conclusions. For instance, they state that age has no appreciable effect on the GFR during pediatric life, yet the only true follow-up study in figure 2 shows a downward trend. So, it is not at all certain that age has no effect.

It may also a bit bold to state that a relevant minority has chronic kidney disease. We do not know whether compensatory mechanisms exist. It would also be too far-stretched to state that living kidney donors have chronic kidney disease on the basis of a lower GFR after donation. Follow-up studies, at least, do not indicate that.

Minor comment

Line 51:  typo. ‘Cystatin’, not ‘Cystatic’.

Reviewer 2 Report

The authors conducted a systemic review on GFR measurements in former ELBW infants based on the data available online by using Pubmed, EMBASE, WofS, and Clinical Trials.Gov databases. Based on the search strategy and results, the authors finally narrowed to a total of 9 studies matching their criteria. For all of the study, the authors finally described a pattern of reduced GFR in former ELBW cases throughout childhood. Though the study followed a strategy to review the online databases, the study failed to prove an exceptional outcome. There are several previous studies show that GFR reduced in former ELBW cases (eg., Pediatr Nephrol. 2005 May; 20 (5): 579-84) and many studies, evidence that there are no changes occurring with GFR rates in ELBW cases throughout their childhood (eg. Acta Paediatr. 2010 Aug; 99 (8): 1192-8). So, with the minimal information and lack of quality assessment, the study did not suitable for publication.